# Let's talk about PFAS: Inconsistent public awareness about PFAS and its sources in the United States

**T. Allen Berthold, Audrey McCrary, Stephanie deVilleneuve** *, **Michael Schramm**

Texas Water Resources Institute, Texas A&M AgriLife, College Station, Texas, United States of America

* s.deVilleneuve@ag.tamu.edu

## Abstract

The presence of per- and polyfluoroalkyl substances (PFAS) in U.S. drinking water has recently garnered significant attention from the media, federal government, and public health professionals. While concerns for PFAS exposure continue to mount, the general public's awareness and knowledge of the contaminant has remained unknown. This exploratory study sought to fill this data gap by administering a nationwide survey in which the awareness of PFAS and community contamination, awareness of PFAS containing products and intentions to change product use, and awareness and concern about PFAS in drinking water were assessed. The results indicated that almost half the respondents had never heard of PFAS and do not know what it is (45.1%). Additionally, 31.6% responded that they had heard of PFAS but do not know what it is. A large portion of respondents (97.4%) also responded that they did not believe their drinking water had been impacted by PFAS. Demographic association did not influence knowledge of PFAS or levels of concern with PFAS in drinking water. The strongest predictor of PFAS awareness was awareness due to known community exposure. The respondents aware of community exposure were more likely to have knowledge of PFAS sources, change their use of items with potential PFAS contamination, and answer that their drinking water sources were also contaminated with PFAS. Based on the received responses, PFAS information and health risks need to be better communicated to the public to help increase awareness. These efforts should also be coordinated between government agencies, utilities, the research community, and other responsible entities to bolster their effectiveness.

**Data Availability Statement:** All raw data files are available from the Zenodo database. https://doi.org/10.5281/zenodo.8132987.

## Introduction

Over the past two decades, per- and polyfluoroalkyl substances (PFAS) have caught the attention of international researchers and governments as a rapidly emerging environmental contaminant. Often referred to by the news media as "forever chemicals," PFAS are a group of thousands of synthetic fluorinated chemical compounds that degrade slowly in the environment due to their extremely strong carbon-fluorine bond [1,2]. Since the late 1940s, PFAS have been produced and utilized in a wide range of industrial processes and consumer

**Funding:** The authors received no specific funding for this work.

**Competing interests:** The authors have declared that no competing interests exist.

products because they are incredibly stable, non-reactive, and hydrophobic [3]. Examples of these products include food packaging, non-stick cookware, household upholstery, personal care products, and cleaning supplies. PFAS are also an integral component of fire extinguishing foams, or aqueous film-forming foams (AFFFs), used frequently in emergency response events and firefighting training activities [3,4].

A result of the extensive sources and high mobility of PFAS, in conjunction with their resistance to degradation, is that they have been bioaccumulating in soil, water, and air over time. This has consequently led to nearly all populations in developed countries having detectable levels of PFAS in their blood serum [5,6]. The widespread human exposure to PFAS can be partially attributed to its growing presence in surface and ground water sources used for public drinking water supplies and private drinking water wells. A recent United States Geological Service study found that at least half of the nation's tap water supply is exposed to some amount of PFAS chemicals [7]. Drinking water is considered one of the dominant routes of exposure to PFAS for populations across the globe, particularly in communities that are near contaminated waters [8–10]. This has caused concern from public health professionals because exposure to PFAS has been linked to negative health effects such as cancer, irregular hormone development, liver damage, weakened immune systems, and reproductive harm [11].

While scientists continue to dissect the complexity of PFAS exposure in humans, the U.S. government's response to regulating PFAS levels in drinking water has, until recently, been limited. In 2016, the United States Environmental Protection Agency (EPA) established the lifetime health advisory level for exposure to perfluorooctanoic acid (PFOA) and perfluorooctane sulfonate (PFOS) types of PFAS from drinking water at 70 ppt [12]. Andrews and Naidenko [13] estimate 0.4–1 million people are exposed to 70 ppt combined PFAS in drinking water systems and as much as 18–80 million or 8%-22% of the US population at 10 ppt. Cadwallader et al. [14] provides slightly higher estimates of 0.93 to 1.96 million people at 70 ppt and national level mean exposures of 4.7–5.2 ppt using Bayesian mixed models and the UCMR3 dataset. In March 2023, the EPA proposed an enforceable National Primary Drinking Water Regulation (NPDWR) to establish Maximum Contaminant Levels for six PFAS variants [15]. The newly proposed NPDWR seeks to reduce the enforceable maximum contaminant levels from 70 ppt to 4 ppt [15]. If finalized, this regulation will help reduce the levels of PFAS in drinking water, monitor for the presence of PFAS, and better notify the public of the levels of PFAS in their local water systems. The EPA has also allocated billions of dollars from the Infrastructure Investment and Jobs Act to improve states' drinking water systems, including the addition of PFAS detection and monitoring. In addition to the United States government's response to the growing threat of PFAS, multiple large corporations have been sued by communities across the country seeking damages and liability for PFAS clean up in municipal water supplies. One of the major corporations in these lawsuits is 3M, who reached a $10.3 billion settlement in June 2023 in which they will pay out money over 13 years to any cities and counties that want to test and clean up PFAS in their public water supplies [16]. Chemical manufacturers Chemours, DuPont, and Corteva also reached a settlement in June 2023 to pay $1.18 billion to remove PFAS from public drinking water systems [17]. Obsekov et al. [18] estimated the financial burden of broader health impacts of PFAS to be between $5.5 and $62.6 billion, rendering regulatory intervention and adoption of alternatives to PFAS the more economically viable alternative to continued PFAS use and exposure.

An increase in awareness of risks of environmental contaminants generally leads to changes in social stigma that spurs political, economic, and regulatory changes [19]. A key factor in the transition from overall awareness to meaningful behavioral and policy changes is the level of concern about the impact of pollutants on human health [20]. Public awareness of numerous environmental contaminants such as asbestos, lead, chlorinated hydrocarbons (e.g., PCBs,

DDT, dioxins), and their effects on health have led to regulatory and voluntary changes in the use of these contaminants in manufactured products [21–23]. The rapidly evolving scientific understanding of impacts of PFAS on human health likely contributes to the unawareness and uncertainty of the general public and slow pace of regulatory intervention [24,25].

Risk perception and avoidance research is primarily derived from public health studies concerned with disease mitigation. For example, social consequences, perceptions, and behavioral changes related to smoking and tobacco consumption have been investigated heavily since the landmark study by Wynder and Graham [26], which was the first to link smoking to lung cancer [27]. More recently, copious studies on COVID-19 risk perception and behavioral adaptations by the public were generated during the global pandemic [28]. The abundance of literature on these and other public health topics over the last century has allowed different frameworks of public perception to be extended into other fields of study, including climate change, pollution, food safety, and even nuclear energy, to name a few [29–32]. In these studies across disciplines, one common theme is concluded from data: awareness, knowledge, and personal experience have a positive effect on behavioral changes. Existing research on public perceptions of PFAS has not fully investigated these factors, therefore the extension of these conclusions to the subject of PFAS is not currently established.

Internationally, concern and behaviors surrounding PFAS exposure have been sparsely studied. In Italy, concern about and perceived risk of detrimental health effects from PFAS exposure were elevated for mothers who had children, a wider social network, higher trust in scientific sources and social media, and were not employed full-time [33]. In Girardi et al. [33], the presence of social networking and trust in scientific information was a key predictor of an increase in subjective knowledge about PFAS. Communities in Australia expressed concern about the uncertainty surrounding PFAS, including its impact on long-term health outcomes and socio-economic impacts of contamination in localized areas [34]. Awareness of PFAS contamination, their magnitude, and potential impacts are still underdeveloped within both government agencies and general populations in Asia [35]. However, a single study focused on culinary preferences in India found 61.9% of the surveyed respondents were not aware of PFAS presence in non-stick cookware and intention to use non-stick cookware declined after information was given about the potential leaching of PFAS from this source [36].

The existing literature on public perceptions of PFAS primarily consists of studies about the experience of residents who have been directly affected by industrial PFAS contamination. Wickham and Shriver [25] found that scientific uncertainty led to mixed messaging from government agencies which increased anxiety and concern around acute PFAS contamination in North Carolina communities. Other stressors in affected communities include uncertainty about PFAS exposure pathways, timing of health effects, and financial burdens from decontamination of sources and medical treatments [37]. Many community members reported hearing about PFAS contamination through local news, neighbors, or incidental interaction with government responses [37]. The few, broader studies on public interaction with information about PFAS have noted an acceleration in published news articles and social media posts within the last decade, with a substantial surge occurring in the last two years [38,39]. However, it is unclear whether this increase in information is equivalent to an increase in awareness and action for the general population, who may not have personal experience with direct PFAS contamination [40]. Studies have not yet characterized the perceptions of PFAS across different communities that comprise the broader U.S. landscape.

Given the gap in research on awareness, concerns, and behaviors related to PFAS for the general public in the U.S., we designed this study to assess the population's: (1) awareness of PFAS and community contamination, (2) awareness of PFAS containing products and

intentions to change product use, and (3) awareness and concern about PFAS in drinking water. The survey was done using a nationally representative sample, so that the following results could be generalized for the broader U.S. population. This work was designed to provide a baseline measurement of these parameters so that the impact of future social and regulatory changes regarding the use of PFAS can be correctly discerned and accurately measured.

## Methods

### Survey instrument

The target population was the general U.S. population aged 18 or older. The survey was distributed in April 2023 through Qualtrics online panels. Panels were continuously sampled until a nationally representative sample was obtained. To approximate a representative sample of the U.S. population, panels were recruited using gender, age, and race/ethnicity quotas. The total sample size was 1,100 respondents and was estimated to be representative of the U.S. population within a ±3% margin of error at the 95% confidence level [41]. A summary of the demographic profile of sample respondents is included in the (S1 Table).

Qualtrics performed quality control checks to ensure response validity, including attention checks, survey duration checks, and IP address checks to prevent duplicate responses. Surveys that failed attention and speed checks, provided invalid answers, or did not meet representative demographic criteria were excluded. The Texas A&M University Institutional Review Board (TAMU IRB) reviewed the study protocol and survey instrument prior to distribution. TAMU IRB deemed the study to be exempt from formal review. Written informed consent was obtained from all participants in the first question of the survey instrument.

To determine awareness of PFAS, respondents were asked if they had heard of PFAS and their level of confidence in their knowledge about it. To assess community exposure, participants were also asked if, to the best of their knowledge, their community had been exposed to PFAS. To assess familiarity with sources of PFAS, survey participants were asked to rate their familiarity with 13 different potential items that might be contaminated with or cause PFAS contamination and their intentions to change use of those items. Although the use of PFAS compounds is much more extensive across industrial and consumer products [42], the items included in this survey were intended to be consistent with item categories currently summarized in U.S. Environmental Protection Agency reports and action plans [43]. To explore awareness and concern about PFAS contamination specifically in drinking water, we asked survey participants for their primary source of drinking water, if their primary source of drinking water had been impacted by PFAS, and their level of concern about PFAS in drinking water. Additionally, respondents were asked to estimate what percentage of the U.S. population they thought had been exposed to PFAS. Questions used for non-demographic variables are included in S2 Table.

### Survey analysis

Although we applied sampling quotas, the returned marginal population levels did not completely match recent national-level statistics. Prior to analysis, individual survey responses were weighted so that marginal proportions of the survey more closely matched national level benchmarks from the 5-year 2021 American Community Survey (ACS) [44] on sex/gender, age group, race/ethnicity, and education level (Table 1). Weights on gender were developed by re-coding "female" and "other" responses as "non-male" because the ACS only provides binary response options for sex. Using this approach, responses from both "female" and "other" respondents have the same marginal weight. Kennedy et al. [45] provide substantial discussion on the treatment of sex and gender in survey adjustment. Due to small subpopulation sample

**Table 1. Unadjusted and adjusted survey profile with target marginal population benchmarks derived from the 2021 American Community Survey [44].**

| Characteristic | Unweighted N | Unweighted % | Target % | Weighted N | Weighted % |
|---|---|---|---|---|---|
| **Age** | | | | | |
| 18:24 | 125 | 11.4 | 11.9 | 130.6 | 11.9 |
| 25:34 | 192 | 17.5 | 17.7 | 195.1 | 17.7 |
| 35:44 | 204 | 18.5 | 16.6 | 183.1 | 16.6 |
| 45:54 | 198 | 18.0 | 16.3 | 179.2 | 16.3 |
| 55:64 | 171 | 15.5 | 16.8 | 184.4 | 16.8 |
| 65+ | 208 | 18.9 | 20.7 | 227.6 | 20.7 |
| No answer | 2 | 0.2 | - | - | - |
| **Education** | | | | | |
| Some high school | 47 | 4.3 | 7.8 | 85.8 | 7.8 |
| High school graduate or GED | 418 | 38.0 | 49.4 | 543.7 | 49.4 |
| Associate's degree | 178 | 16.2 | 8.3 | 91.3 | 8.3 |
| Bachelor's degree | 246 | 22.4 | 19.4 | 213.7 | 19.4 |
| Master's degree | 132 | 12.0 | 8.3 | 91.3 | 8.3 |
| Doctorate or terminal degree | 28 | 2.5 | 1.3 | 14.7 | 1.3 |
| Other | 40 | 3.6 | 5.4 | 59.5 | 5.4 |
| No answer | 11 | 1.0 | - | - | - |
| **Race/Ethnicity** | | | | | |
| White | 723 | 65.7 | 62.4 | 686.3 | 62.4 |
| Non-white | 373 | 33.9 | 37.6 | 413.7 | 37.6 |
| No answer | 4 | 0.4 | - | - | - |
| **Sex/Gender** | | | | | |
| Male | 529 | 48.1 | 49.0 | 539.1 | 49.0 |
| Not Male | 569 | 51.7 | 51.0 | 560.9 | 51.0 |
| No answer | 2 | 0.2 | - | - | - |

sizes within the race/ethnicity variable, race/ethnicity were recoded as white or Caucasian and non-white categories. Weights were developed by poststratification raking using the American National Election Study (ANES) weighting algorithm implemented in the *anesrake* R package [46,47].

To explore factors associated with an individual's understanding of PFAS, two different proportional odds models [48] were developed relating: 1) self-described knowledge of PFAS (4 responses ranging from "I've never heard of it, and don't know what it is" to "I'm confident I know what it is"); and 2) awareness of potential sources of PFAS (5 responses ranging from "Not at all familiar" to "Extremely familiar"; to sex/gender, age, race/ethnicity, education, and awareness of community exposure to PFAS. To explore factors associated with intended behavior change, a proportional odds model was developed relating intention to change use of items associated with PFAS (5 responses ranging from "Will never change" to "Have already changed") to the same dependent variables.

Additionally, the probability that an individual was aware of PFAS impacting their drinking water was explored using a logistic regression model relating awareness of PFAS contamination in drinking water (dummy variable) to sex/gender, age, race/ethnicity, education, drinking water source, and awareness of community exposure to PFAS. A final model evaluated the factors associated with an individual's level of concern about PFAS in their drinking water using a proportional odds model to fit level of concern (5 responses ranging from "Not at all concerned" to "Extremely concerned") to sex/gender, age, race/ethnicity, education, drinking water source, and awareness of PFAS contamination in drinking water.

Model results are presented as odds-ratios (with approximate p-values calculated by comparing the t-value against the standard normal distribution). Marginal effects are also presented as population-level predicted probabilities for a given predictor estimated using observed values [49]. Confidence intervals (95%) were derived using a parametric bootstrap as implemented in the *svyEffects* R package [50]. All models were fit using the *survey* package in R version 4.2.1 [51,52].

## Results

Most respondents reported no knowledge of (41.1%) or were unsure (47.4%) if their community had been exposed to PFAS (Table 2). Only 11.5% responded that they knew their community has been exposed to PFAS. When asked to describe knowledge level about PFAS, 45.1% responded that they have never heard of it and do not know what it is. An additional 31.6% responded they have heard of PFAS, but do not know what PFAS is. On average, respondents estimated that 54.2% of the U.S. population had been exposed to PFAS.

Most individuals said they use unfiltered (27.9%) or filtered (37.6%) tap water as their main source of drinking water. A large majority of people responded that, to their knowledge, their drinking water had not been impacted by PFAS (97.4%). When asked about their level of concern about PFAS in drinking water, 23.1% had no concerns, 17.8% and 24.3% were slightly or moderately concerned. Fewer people responded that they were extremely concerned (15.7%) or very concerned (19.1%) about PFAS in their drinking water.

**Table 2. Population level estimates of responses for PFAS knowledge, awareness of community exposure, sources of drinking water, awareness of drinking water contamination, and concern about drinking water contamination.**

| Question | Percent Response, (SE) |
|---|---|
| **What is your main source of drinking water?** | |
| Unfiltered tap water | 27.9 (1.5) |
| Filtered tap water | 37.6 (1.6) |
| Bottled/prepackaged water | 34.2 (1.6) |
| Other | 0.3 (0.1) |
| **To your knowledge, has your primary source of drinking water been impacted by PFAS?** | |
| No | 97.4 (0.5) |
| Yes | 2.6 (0.5) |
| **How concerned are you about PFAS in your drinking water?** | |
| Not at all concerned | 23.1 (1.4) |
| Slightly concerned | 17.8 (1.2) |
| Moderately concerned | 24.3 (1.4) |
| Very concerned | 19.1 (1.3) |
| Extremely concerned | 15.7 (1.2) |
| **To your knowledge, has your community been exposed to PFAS?** | |
| Yes | 11.5 (1.0) |
| No | 41.1 (1.6) |
| Not sure | 47.4 (1.6) |
| **How would you describe your knowledge about PFAS as an environmental contaminant?** | |
| I've never heard of it, and don't know what it is | 45.1 (1.6) |
| I've heard of it or seen it somewhere, but don't know what it is | 31.6 (1.5) |
| I think I know what it is | 17.2 (1.2) |
| I'm confident I know what it is | 6.2 (0.8) |

**Table 3. Population-level estimates of percent responses to awareness of different potential sources of PFAS contamination.**

| Sources | Percent Response | | | | |
|---|---|---|---|---|---|
| | Not at all familiar[1] | Slightly familiar[1] | Moderately familiar[1] | Very familiar[1] | Extremely familiar[1] |
| Drinking water | 45.8 (1.6) | 19.7 (1.3) | 17.1 (1.2) | 9.2 (0.9) | 8.2 (0.9) |
| Waterways near waste disposal sites | 45.2 (1.6) | 18.3 (1.3) | 20.0 (1.3) | 10.7 (1.0) | 5.9 (0.8) |
| Soils near waste disposal sites | 46.3 (1.6) | 20.2 (1.3) | 17.5 (1.2) | 10.1 (0.9) | 5.9 (0.8) |
| Dairy products | 51.1 (1.6) | 16.3 (1.2) | 15.3 (1.2) | 10.0 (0.9) | 7.3 (0.8) |
| Fresh produce | 50.3 (1.6) | 14.5 (1.1) | 16.1 (1.2) | 11.7 (1.0) | 7.4 (0.8) |
| Freshwater fish | 48.7 (1.6) | 16.4 (1.2) | 17.6 (1.2) | 11.4 (1.0) | 5.9 (0.7) |
| Seafood | 48.8 (1.6) | 15.3 (1.2) | 17.7 (1.2) | 9.9 (1.0) | 8.3 (0.9) |
| Food packaging | 48.1 (1.6) | 16.2 (1.2) | 16.9 (1.2) | 11.8 (1.0) | 7.0 (0.8) |
| Non-stick cookware | 47.0 (1.6) | 16.9 (1.2) | 16.8 (1.2) | 12.8 (1.1) | 6.5 (0.8) |
| Personal hygiene products | 46.6 (1.6) | 14.7 (1.1) | 18.2 (1.3) | 12.6 (1.0) | 8.0 (0.9) |
| Household products | 45.0 (1.6) | 16.0 (1.2) | 18.7 (1.3) | 12.3 (1.0) | 7.9 (0.9) |
| Fire extinguishing foam | 50.7 (1.6) | 14.7 (1.1) | 15.8 (1.2) | 11.7 (1.0) | 7.1 (0.9) |
| Fertilizers from wastewater plants | 45.9 (1.6) | 17.2 (1.2) | 17.0 (1.2) | 11.9 (1.0) | 8.1 (0.9) |

[1]Percent Responses (Standard Error).

On average, 47.6% (SE = 0.6%) of respondents were "Not at all familiar" with potential sources of PFAS included in the survey (Table 3). The probability of response decreased for increasing levels of familiarity across all potential PFAS sources with only 7.2% (SE = 0.3%) of respondents, on average across all sources, responded "Extremely familiar."

Individual intention to change product usage due to PFAS contamination was less certain. Across all items, most individuals responded, "Not sure" (Mean = 24.6%, SE = 0.9%) or "Might change" (Mean = 27.9%, SE = 0.4% Table 4). While the mean response rate across product categories for individuals that have already changed product use was only 11.8% (SE = 0.7%), the drinking water item stood out with 18.9% (SE = 1.3%) of respondents indicating they have already changed use of drinking water.

**Table 4. Population level estimates of percent responses rating intention to change products because of potential for PFAS contamination.**

| Sources | Percent Response | | | | |
|---|---|---|---|---|---|
| | Will never change[1] | Not sure[1] | Might change[1] | Planning to change[1] | Have already changed[1] |
| Drinking water | 15.8 (1.2) | 18.4 (1.3) | 26.5 (1.4) | 20.5 (1.3) | 18.9 (1.3) |
| Waterways near waste disposal sites | 13.9 (1.1) | 26.7 (1.5) | 27.9 (1.4) | 20.5 (1.3) | 11.0 (1.0) |
| Soils near waste disposal sites | 15.7 (1.2) | 28.1 (1.5) | 26.4 (1.4) | 19.8 (1.3) | 10.0 (1.0) |
| Dairy products | 17.8 (1.3) | 22.8 (1.4) | 29.8 (1.5) | 18.6 (1.2) | 11.0 (1.0) |
| Fresh produce | 18.4 (1.3) | 22.5 (1.4) | 28.8 (1.5) | 19.9 (1.3) | 10.4 (1.0) |
| Freshwater fish | 17.3 (1.2) | 25.2 (1.4) | 29.3 (1.5) | 17.9 (1.2) | 10.4 (1.0) |
| Seafood | 18.1 (1.3) | 24.6 (1.4) | 28.0 (1.4) | 19.7 (1.3) | 9.6 (0.9) |
| Food packaging | 15.7 (1.2) | 23.6 (1.4) | 27.7 (1.4) | 21.0 (1.3) | 11.9 (1.0) |
| Non-stick cookware | 15.0 (1.2) | 23.2 (1.4) | 27.9 (1.4) | 18.8 (1.3) | 15.0 (1.1) |
| Personal hygiene products | 15.4 (1.2) | 23.8 (1.4) | 28.9 (1.5) | 20.2 (1.3) | 11.6 (1.0) |
| Household products | 15.1 (1.2) | 22.0 (1.4) | 29.8 (1.5) | 21.5 (1.3) | 11.5 (1.0) |
| Fire extinguishing foam | 17.4 (1.2) | 29.7 (1.5) | 25.7 (1.4) | 16.6 (1.2) | 10.7 (1.0) |
| Fertilizers from wastewater plants | 14.2 (1.1) | 28.6 (1.4) | 26.0 (1.4) | 19.5 (1.3) | 11.7 (1.0) |

[1]Percent Responses (Standard Error).

## Factors associated with self-described PFAS knowledge and products

We did not find evidence for any association between sex, race/ethnicity, or education with self-described knowledge about PFAS (Table 5). With the sample size used in the current study we were not able to incorporate the sub-populations and develop a model that would

**Table 5. Odds ratios, confidence intervals, and approximate p-values from (Model 1) the proportional odds model relating covariates to self-described knowledge levels about PFAS; (Model 2) the logistic regression model relating covariates with awareness of drinking water contamination from PFAS; and (Model 3) the proportional odds model relating covariates with levels of concerns about PFAS in drinking water.**

| Characteristic | Model 1: Knowledge level of PFAS | | | Model 2: Knowledge of drinking water contamination | | | Model 3: Level of concern about PFAS | | |
|---|---|---|---|---|---|---|---|---|---|
| | OR[1] | 95% CI[1] | p-value | OR[1] | 95% CI[1] | p-value | OR[1] | 95% CI[1] | p-value |
| **Sex/Gender** | | | | | | | | | |
| Male | — | — | | — | — | | — | — | |
| Female | 0.99 | 0.76, 1.29 | >0.9 | 0.66 | 0.25, 1.75 | 0.4 | 1.08 | 0.84, 1.38 | 0.5 |
| Other | 0.88 | 0.19, 3.38 | 0.8 | 0.00 | 0.00, 0.00 | <0.001 | 0.67 | 0.26, 1.77 | 0.4 |
| **Age** | | | | | | | | | |
| 18:24 | — | — | | — | — | | — | — | |
| 25:34 | 1.38 | 0.87, 2.18 | 0.2 | 1.30 | 0.12, 14.0 | 0.8 | 0.93 | 0.63, 1.39 | 0.7 |
| 35:44 | 1.63* | 1.03, 2.59 | 0.039 | 0.72 | 0.07, 7.45 | 0.8 | 1.21 | 0.80, 1.82 | 0.4 |
| 45:54 | 1.03 | 0.64, 1.67 | 0.9 | 1.64 | 0.13, 21.2 | 0.7 | 1.30 | 0.86, 1.96 | 0.2 |
| 55:64 | 1.09 | 0.66, 1.80 | 0.7 | 4.47 | 0.29, 69.7 | 0.3 | 0.80 | 0.50, 1.28 | 0.3 |
| 65+ | 0.94 | 0.57, 1.57 | 0.8 | 0.80 | 0.06, 10.1 | 0.9 | 0.74 | 0.47, 1.16 | 0.2 |
| **Race/Ethnicity** | | | | | | | | | |
| White | — | — | | — | — | | — | — | |
| Non-white | 0.95 | 0.71, 1.27 | 0.7 | 0.58 | 0.19, 1.72 | 0.3 | 1.04 | 0.78, 1.39 | 0.8 |
| **Education** | | | | | | | | | |
| Some high school | — | — | | — | — | | — | — | |
| High school/GED | 0.65 | 0.34, 1.23 | 0.2 | 0.74 | 0.05, 10.7 | 0.8 | 0.67 | 0.34, 1.30 | 0.2 |
| Associate's degree | 0.89 | 0.46, 1.76 | 0.7 | 0.90 | 0.05, 15.0 | >0.9 | 0.54 | 0.27, 1.09 | 0.084 |
| Bachelor's degree | 0.98 | 0.50, 1.92 | >0.9 | 1.46 | 0.13, 17.1 | 0.8 | 0.67 | 0.34, 1.35 | 0.3 |
| Master's degree | 1.04 | 0.51, 2.12 | >0.9 | 1.54 | 0.14, 16.5 | 0.7 | 1.01 | 0.49, 2.08 | >0.9 |
| Doctorate or terminal degree | 1.38 | 0.57, 3.32 | 0.6 | 1.61 | 0.08, 34.2 | 0.8 | 1.05 | 0.39, 2.84 | >0.9 |
| Other | 1.56 | 0.61, 3.94 | 0.4 | 2.20 | 0.15, 31.3 | 0.6 | 0.93 | 0.39, 2.20 | 0.9 |
| **Community PFAS Exposure** | | | | | | | | | |
| Yes | — | — | | — | — | | | | |
| No | 0.28*** | 0.19, 0.41 | <0.001 | 0.01*** | 0.00, 0.09 | <0.001 | | | |
| Not Sure | 0.23*** | 0.16, 0.35 | <0.001 | 0.12** | 0.03, 0.49 | 0.003 | | | |
| **Drinking Water Source** | | | | | | | | | |
| Unfiltered tap water | | | | — | — | | — | — | |
| Filtered tap water | | | | 0.28* | 0.10, 0.79 | 0.016 | 1.28 | 0.95, 1.73 | 0.10 |
| Bottled/prepackaged water | | | | 0.44 | 0.17, 1.19 | 0.11 | 1.44* | 1.05, 1.99 | 0.025 |
| Other | | | | 0.00*** | 0.00, 0.00 | <0.001 | 1.73 | 0.69, 4.34 | 0.2 |
| **Drinking Contaminated by PFAS** | | | | | | | | | |
| No | | | | | | | — | — | |
| Yes | | | | | | | 4.27 | 2.20, 8.31 | <0.001 |

[1] OR = Odds Ratio, CI = Confidence Interval.

*p<0.05

**p<0.01

***p<0.001.

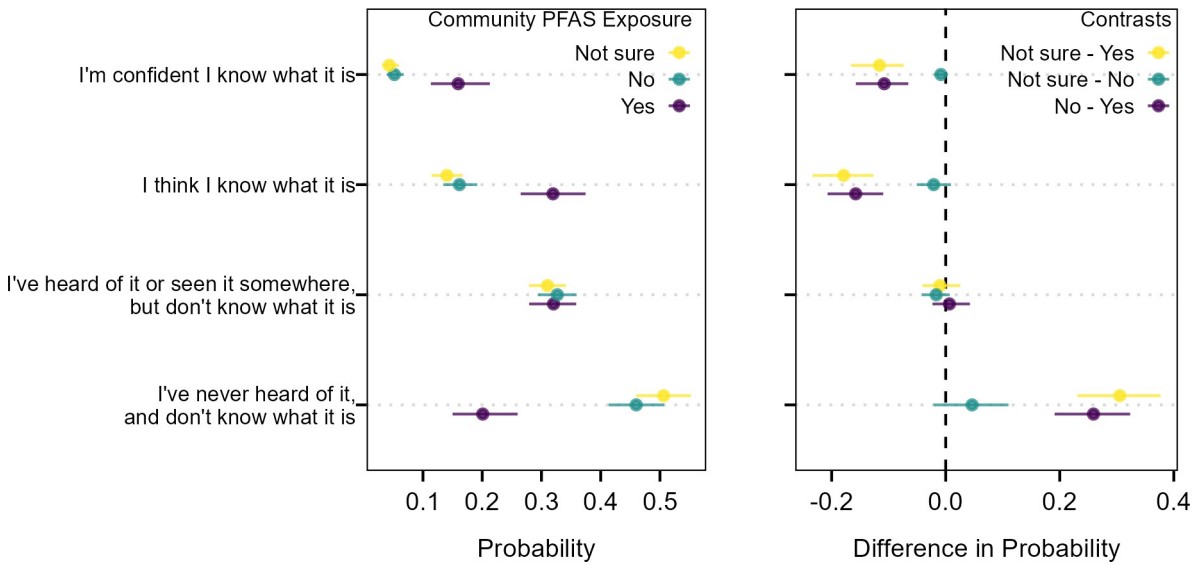

**Fig 1. Average marginal effects (left) and contrasted effects (right) of awareness of community PFAS exposure on self-assessed knowledge of PFAS.** Horizontal lines indicate the 95% confidence intervals of the marginal predicted probabilities and contrasts in marginal predicted probabilities across the population.

converge. There was not strong evidence for the influence of age on PFAS knowledge among most of the age brackets (Table 5). However, there is evidence to support that individuals in the 35:44 age bracket will respond with a higher self-assessed knowledge level (OR = 1.63, p = 0.039; Table 5) than someone in the reference bracket (18:24). There was also strong evidence that people aware of PFAS exposure in their communities self-report higher levels of knowledge about PFAS. People aware of PFAS exposure in their communities are 3.57 times and 4.35 times more likely to respond with a higher self-assessed knowledge level than those responding "No" (OR = 0.28, p < 0.001; Table 5) or "Not sure" (OR = 0.23, p <0.001; Table 5) to awareness of PFAS contamination in their communities.

There was no difference in marginal predicted response probabilities between people that were unsure if their community had been exposed to PFAS or said their community had not been exposed to PFAS for responses to self-assessed knowledge about PFAS (Fig 1). People that were aware that their community had been exposed to PFAS had between a 10.8% to 11.6% higher probability of responding they were confident of their knowledge of PFAS compared to the remaining groups. They also had a 15.8% to 17.9% higher probability of responding they thought that they knew what PFAS was compared to the remaining groups. Conversely, someone that is aware of community PFAS exposure was much less likely to respond that they had never heard of PFAS and did not know what it was (20%) compared to those that said their community has not been exposed (46%) or did not know (51%).

Awareness of community PFAS exposure also shows strong associations with familiarity of potential PFAS sources and intentions to change use of items with potential for PFAS contamination (Figs 2 and 3). On average, 46% of respondents that were unaware and 46.5% of those that were not sure if their communities were contaminated by PFAS responded they were "not familiar at all" with specific sources of PFAS contamination. This decreased to averages of 6% and 4% for the "extremely familiar" response. On average, people that stated their communities were contaminated by PFAS had a lower probability (18%) of responding that they were "not familiar at all" and higher probability (20%) of being "extremely familiar" with PFAS sources compared to the other two groups.

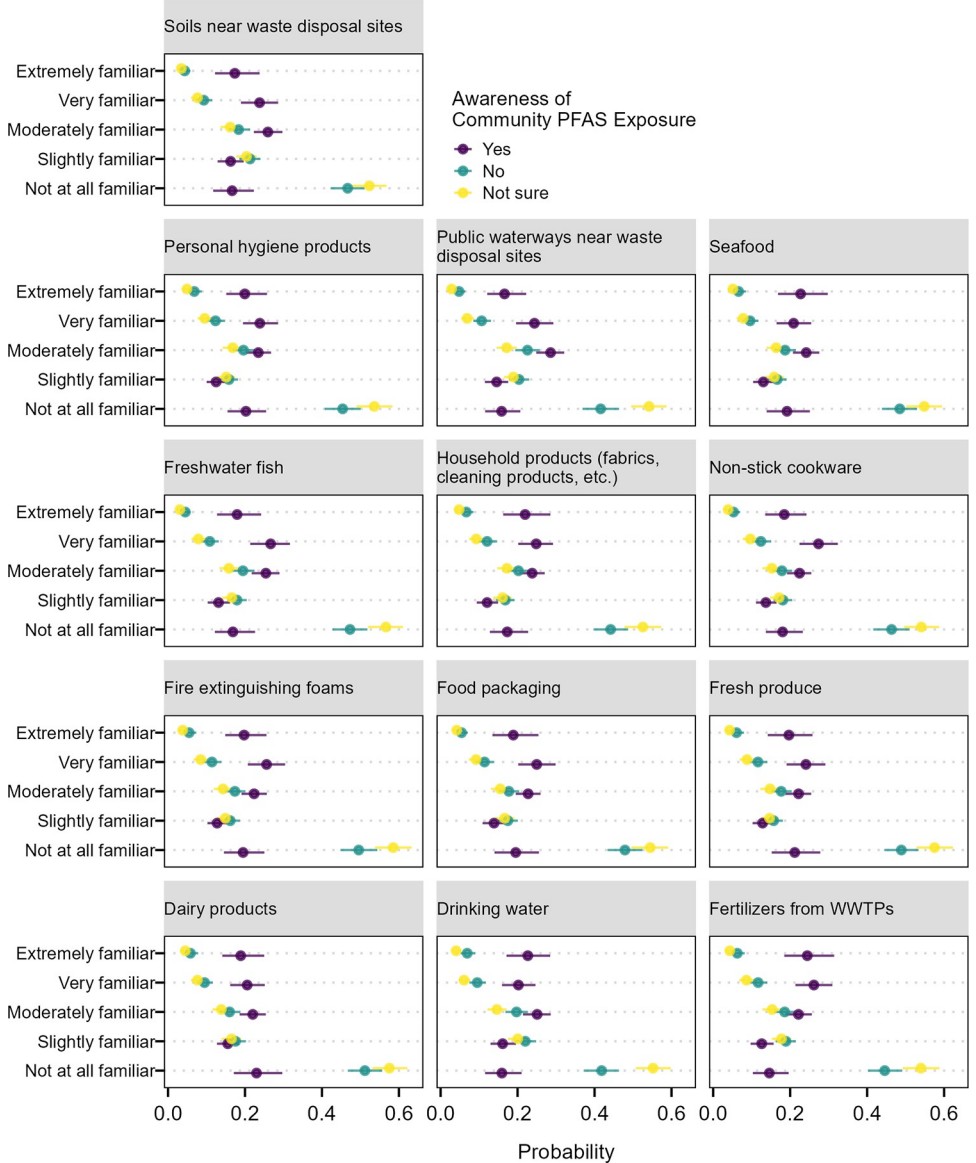

**Fig 2. Average marginal effects of awareness of community PFAS exposure on the response probability for familiarity with different products associated with PFAS contamination.** Horizontal lines are the 95% confidence intervals of the marginal predicted probability across the population.

Respondents who answered no or were unsure of community PFAS contamination were on average more likely to say they will never change their use of items (17% and 17%) compared to those aware of PFAS contamination in their community (8% Fig 3). Those aware of community PFAS contamination were also more likely on average to have already changed use of items (22%) relative to the other two groups (11% for the "no" group and 10% for the "unsure" group).

## PFAS and drinking water contamination

There was not strong evidence that age, race, or education are predictive of an individual's awareness of PFAS contamination in their drinking water (Table 5). There was some evidence

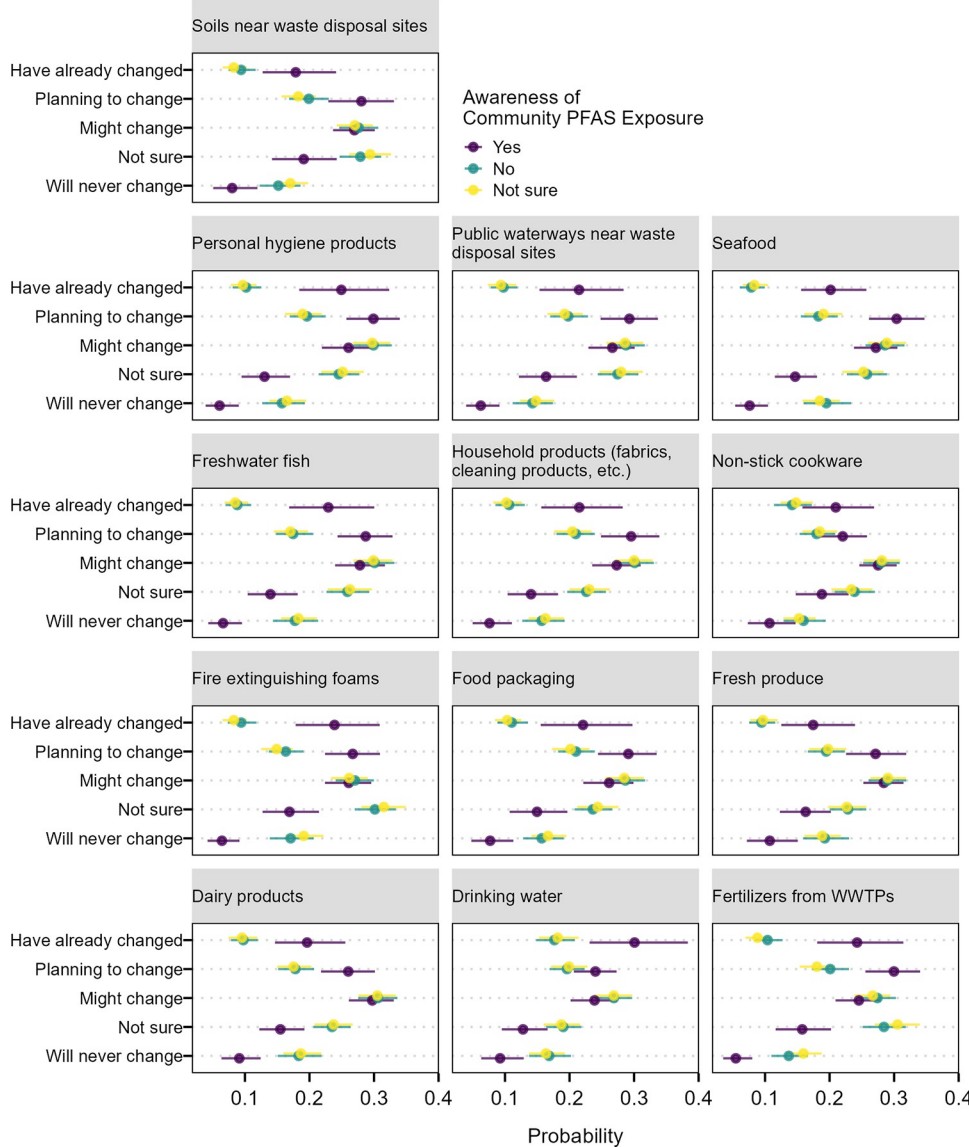

**Fig 3. Average marginal effects of awareness of community PFAS exposure on the response probability for intention to change use of different products associated with PFAS contamination.** Horizontal lines are the 95% confidence intervals of the marginal predicted probability across the population.

supporting correlations with the gender variable, with individuals identifying as other having 100% lower odds of responding that they know that their water is contaminated with PFAS (OR = 0, p < 0.001 Table 5) than individuals identifying male. There was strong evidence that awareness of community PFAS exposure was associated with knowledge that drinking water sources were contaminated with PFAS. The odds that an individual aware of community PFAS exposure indicated their drinking water was contaminated with PFAS was 100 times greater than those that responded they were unaware of community PFAS exposure (OR = 0.01, p < 0.001 Table 5) and 8.3 times lower than those that were uncertain of PFAS exposure (OR = 0.12, p = 0.003 Table 5). There was also evidence for associations between the types of drinking water sources used by an individual and their awareness of their drinking water being contaminated by PFAS. Individuals with unfiltered tap water had 3.57 times the

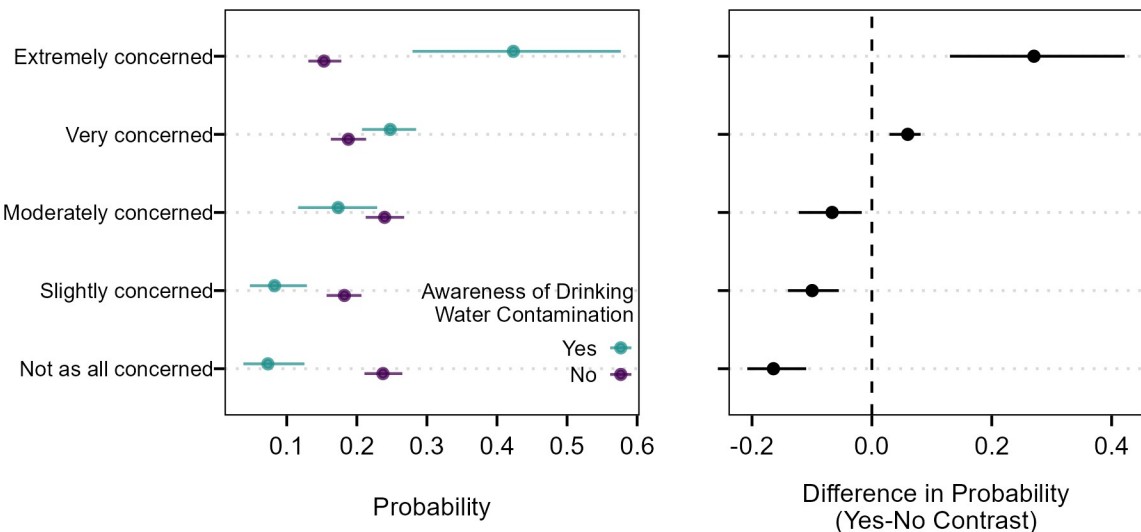

**Fig 4. Average marginal effects (left) and contrasts in effects (right) of awareness of drinking water PFAS contamination on level of concerns about PFAS contamination in drinking water.** Horizontal lines indicate the 95% confidence intervals of the marginal predicted probabilities and contrasts in marginal predicted probabilities across the population.

odds of being aware that their drinking water was contaminated than those with filtered tap water (OR = 0.28, p = 0.016 Table 5).

There was not strong evidence that sex/gender, age, race/ethnicity, or education were associated with levels of concern with PFAS contamination in drinking water (Table 5). There was some evidence of associations between source of drinking water and levels of concern about PFAS contamination with users of bottled/prepackaged water having 1.44 times the odds of higher levels of concern about PFAS contamination in drinking water compared to those that use unfiltered tap water (p = 0.025 Table 5). There was strong evidence of associations between awareness of PFAS contamination in drinking water and concern about PFAS in drinking water. An individual that was aware of PFAS contamination in their drinking water had 4.27 times higher odds of reporting a higher level of concern about PFAS contamination that someone that was unaware of contamination (p = <0.001 Table 5). Marginal predicted probabilities show that an individual aware of PFAS contamination in their drinking water had a 27.0% higher probability of being "Extremely concerned" about PFAS contamination and 6.0% higher probability of being "Very concerned" (Fig 4). Conversely, individuals who said their drinking water was not contaminated by PFAS were more likely to respond that they were "Not at all concerned", "Slightly concerned", or "Moderately concerned".

## Discussion

To our knowledge, this study was the first measure of awareness of PFAS within the general U. S. population. Overall, only about half of the respondents stated they were aware of PFAS as an environmental contaminant, while 76% of respondents stated they did not know what PFAS are. Despite these gaps, most respondents stated that they had some level of concern about PFAS in their drinking water. Those who were the most concerned with PFAS contaminating their drinking water were also those who indicated their primary source of drinking water had been contaminated. Community exposure appears to be the strongest predicting factor regarding the level of public knowledge and awareness of PFAS and its sources.

Individuals who responded that they were aware of PFAS contamination in their community generally reported higher perceived knowledge of PFAS as an environmental

contaminant. This relationship is consistent with findings by Liu and Yang [40], who concluded that an increase in perceived personal relevance of PFAS boosts the information-seeking behaviors of individuals to reach a level of sufficient, or useful, knowledge. However, in other studies, individuals who experienced acute contamination in their communities still expressed uncertainty about the practical aspects of PFAS knowledge, including pathways of exposure, health risks, and potential mitigation strategies [25,37]. While improved awareness and knowledge of PFAS is logical for communities that have faced acute contamination, it appears that the flood of information about PFAS as a contaminant may not translate into practical understanding of PFAS exposure and the health implications in everyday life. Our results show that awareness and knowledge of PFAS is underdeveloped in the general population, which could mean the broader public either has not reached the threshold where PFAS is relevant enough to prompt information-seeking behaviors, or existing communications about PFAS are not translating the personal relevance of the abundance of exposure pathways or the long-term health implications effectively.

In communities exposed to industrial PFAS contamination, uncertainty about the chemical stems from the conflicting information presented by local government, state agencies, and PFAS manufacturers responding to the situation [25,37]. Likewise, the broader population also faces uncertainty about where to find practical information for PFAS and details about their impact on public health [38]. Currently, there is no definitive answer for who is responsible for communicating the risks of PFAS to the general public, which can make individuals and communities feel isolated from the scientific and regulatory discourse [38]. Distrust in political institutions and the opaque use of jargon by PFAS manufacturers limits the effectiveness of their involvement in public outreach [25]. Some findings suggest scientific institutions, such as environmental agencies, universities, and national or state research institutes, may be the best authority for communicating about PFAS because they are more trusted to provide timely and actionable information [53]. Conflicting information about PFAS will continue to stifle public awareness, and in turn, limit effective action and regulatory policy until more cohesive and decisive messaging is adopted. A discussion about which institutions have responsibility for PFAS messaging and honest feedback about the effects of PFAS could help create a unified communication strategy, so that social discourse about PFAS is unambiguous, honest, and reliable–to the benefit of the general public.

Regardless of who sends the message, entities who communicate to the public must be clear about what is known about PFAS so individuals, health professionals, and communities can make educated decisions to minimize exposure [54]. On average, nearly half of survey respondents were completely unfamiliar with various consumer products as sources of PFAS. Again, those who were aware that their community had been affected by PFAS were more likely to say they had greater familiarity with everyday sources of PFAS. A survey by Shin et al. [20] found that chemicals in consumer products were the most common concern related to environmental health risk. Our survey revealed that while 55% of the population may have heard of PFAS, just 23% felt they understood PFAS as an environmental contaminant, meaning most of the population did not know about its uses, risks, or extent as a chemical present in consumer products. Similarly, Dong and Yang [53] found that respondents felt they had just a quarter of the sufficient knowledge needed to make informed decisions about the risks of PFAS to their personal health. Insufficient knowledge is a clear detriment for the adoption of behaviors that reduce personal risk [55]. However, knowledge must have subjective context to affect behavioral intentions and outcomes of individuals [56]. The knowledge presented about PFAS in products must not just be broadcast, but also be unambiguous, relevant, and actionable, so that more consumers can make informed decisions about their level of exposure.

The intention to change the use of consumer products containing PFAS was (again) closely linked to awareness of PFAS exposure within the respondent's community and familiarity with PFAS sources. Individuals who are more aware of the risks of exposure may be more motivated to seek information and act on it [57,58]. Other literature has shown perceived social responsibility, which can be influenced through direct primary contacts (friends, family, coworkers) or mass media, to be a critical factor in risk avoidant behaviors at an individual level as well [28,31,59]. However, there may be other factors acting as barriers to action for those not intending to change usage habits, such as the costs associated with avoiding PFAS in drinking water and food products (such as replacing cookware or installing filters) and the perceived efficacy of remediation [58]. Citizens in communities with severe exposure to PFAS contamination have cited lack of resources, including financial and technical assistance, as a limiting factor for avoiding PFAS-contaminated water [37]. Increasing awareness to encourage knowledge-seeking behavior and spur changes in intention may be helpful, but these intentions may not be achievable for many individuals, especially those in low-income populations. More research into the efficacy of financial and technical assistance to address PFAS in both acutely contaminated communities and the broader population may help uncover effective solutions. Furthermore, regulatory intervention to reduce the baseline environmental presence of PFAS may help narrow the gap between contamination and remediation.

In our study, those who responded that they were aware their community had been exposed to PFAS were also significantly more likely to have knowledge of contamination in drinking water. Though only 2.6% of respondents believed their primary source of drinking water had been exposed to PFAS, the presence of PFAS in drinking water in the U.S. is quite extensive, with an estimated 45% of all drinking water samples containing at least one type of PFAS, according to a recent study by Smalling et al. [7]. In communities near contaminated sites, approximately 75% of PFAS exposure comes from drinking water alone [8]. Our results indicate that awareness of community PFAS contamination may largely be attributed to known contamination of drinking water supplies, even though dietary exposure is the major contributor of population PFAS exposure in the U.S. [60,61]. The increased awareness of PFAS in communities affected by acute drinking water contamination may be attributed to heightened local media attention and warnings from drinking water suppliers and city governments. However, the results also show that 20% of the respondents who are aware of PFAS exposure in their drinking water also described themselves as having limited knowledge of PFAS. Therefore, it is important to emphasize education efforts in areas where community exposure is high. Dietary exposure to PFAS, as well as exposure through dust inhalation and consumer goods, is likely unreported at the community level because it is rarely traced to point source contamination and the innumerable pathways of exposure leave the responsibility of communicating the risks uncertain. If known community exposure is the key to PFAS knowledge and awareness, municipalities and private water suppliers should prioritize funds to test for PFAS contamination in drinking water systems. This would enable these entities to provide the public with accurate, real-time data that creates a personal link between the consumers and PFAS exposure, likely leading them to want to know more. Manufacturers of PFAS products should also clearly label their products with the presence of the chemical, so that consumers can be aware of their exposure frequency and make adjustments in use as desired.

While the results indicate there are no strong connections between individual PFAS awareness in drinking water and demographics such as race, education level, and age, certain groups are more likely to be exposed to PFAS from contaminated drinking water. For example, Liddie [62] found community water systems with higher concentrations of PFAS were more likely to serve greater proportions of Hispanic/Latino and non-Hispanic black populations. A study conducted at a PFAS contaminated water supply in Paulsboro, New Jersey found that

perfluorononanoic acid (PFNA) blood serum levels were higher in older residents compared to younger, and higher in males compared to females [63]. Other studies support the findings that males typically have higher concentrations of PFAS in their blood serum than females [5,44,64,65] and that PFAS exposure from drinking water increases in magnitude with age [66]. Lower concentrations of PFAS in female serum is likely due to elimination from menstrual blood loss and PFAS transfer during breastfeeding [67]. Greater concentrations in older populations may be due to higher cumulative exposure or changes in susceptibility [66]. Due to the lack of PFAS awareness from drinking water across all demographic groups, entities aimed at increasing public awareness should spend more resources targeting groups that are likely to be exposed to greater concentrations of PFAS from drinking water such as minorities, males, and older populations. While increasing the awareness of all groups is ideal, prioritizing demographics with higher exposure may help change the behaviors of those who are the most vulnerable to significant health risks.

Within the study design and subsequent dataset there were a few limitations that could have influenced certain outcomes. One limitation is that race/ethnicity sub-populations effects cannot be excluded because race/ethnicity was collapsed into "white" and "non-white" categories. For example, all the respondents identifying as "Native Hawaiian or Other Pacific Islander" (unweighted n = 3) responded that they had never heard of or knew what PFAS were. Some sub-populations might be more likely to answer that they have less knowledge about PFAS. Another limitation was the absence of geographic targets or weighting in the analysis. This can skew the results to be over or under targeting areas with PFAS contamination relative to the overall population. Additionally, the use of an online survey instrument creates inherent bias against portions of the U.S. population who may not have internet access. An estimated 7%, or nearly 23 million Americans, do not use the internet [68]. Therefore, the sample obtained from this survey may not be generalizable to this portion of the population that is not accessible via internet-based survey instruments.

## Conclusion

This study used an exploratory public survey to identify large discrepancies in awareness about PFAS, its sources, and the adoption of behavioral change to avoid PFAS exposure among the U.S. population. Through the analysis it became clear that greater PFAS awareness, knowledge, and willingness to change behavior is associated with communities that have known PFAS drinking water contamination. The inconsistent public awareness about PFAS indicates that improved efforts in educating the public still need to be undertaken by the U.S. government, utilities, universities, state extension services, and other scientific institutions with high public trust. While there is widely available information regarding PFAS sources and negative health effects, an overwhelming majority of the U.S. population are still completely unaware of what PFAS are, even if they have heard of them. The rapidly evolving scientific understanding of PFAS has led to uncertain messaging to the public which can impact their overall awareness. The widespread nature of the presence of PFAS in humans is also not common knowledge amongst the American people as only a small fraction of respondents were aware that their primary source of drinking water had been impacted, despite studies that have determined otherwise. This study builds on a growing body of evidence that improved messaging and communication about PFAS, its sources, and its health risks are needed. As scientific understanding of the health impacts and scope of PFAS exposure increases, coordinated efforts are also required among government agencies, the research community, and utilities to develop and evaluate the effectiveness of public messaging efforts.

## Supporting information

**S1 Table. Unadjusted demographic characteristics of all survey respondents.**
(PDF)

**S2 Table. Questions and response options used in survey instrument.**
(PDF)

## Acknowledgments

The authors would like to thank Ed Rhodes, of the Texas Water Resources Institute, for his insight regarding the draft of the manuscript.

## Author Contributions

**Conceptualization:** T. Allen Berthold.

**Data curation:** Audrey McCrary.

**Formal analysis:** Michael Schramm.

**Investigation:** Audrey McCrary, Stephanie deVilleneuve.

**Methodology:** T. Allen Berthold, Audrey McCrary, Stephanie deVilleneuve, Michael Schramm.

**Project administration:** T. Allen Berthold.

**Software:** Michael Schramm.

**Visualization:** Michael Schramm.

**Writing – original draft:** Audrey McCrary, Stephanie deVilleneuve, Michael Schramm.

**Writing – review & editing:** T. Allen Berthold.

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
