## [Decision Letter · Decision Letter 0]

15 Aug 2023

PONE-D-23-21851Let's talk about PFAS: Inconsistent public awareness about PFAS and its sources in the United StatesPLOS ONE

Dear Dr. deVilleneuve,

Thank you for submitting your manuscript to PLOS ONE. After careful consideration, we feel that it has merit but does not fully meet PLOS ONE’s publication criteria as it currently stands. Therefore, we invite you to submit a revised version of the manuscript that addresses the points raised during the review process.

We look forward to receiving your revised manuscript.

Kind regards,

Linglin Xie

Academic Editor

PLOS ONE

Journal Requirements:

3. Please include a copy of Table 6 which you refer to in your text on page 18.

Reviewers' comments:

Reviewer's Responses to Questions

**Comments to the Author**

1. Is the manuscript technically sound, and do the data support the conclusions?

Reviewer #1: Yes

Reviewer #2: Yes

2. Has the statistical analysis been performed appropriately and rigorously? 

Reviewer #1: No

Reviewer #2: N/A

3. Have the authors made all data underlying the findings in their manuscript fully available?

Reviewer #1: Yes

Reviewer #2: Yes

4. Is the manuscript presented in an intelligible fashion and written in standard English?

Reviewer #1: Yes

Reviewer #2: Yes

5. Review Comments to the Author

Reviewer #1: The article has measured the awareness of PFAS within the general US population. However, there are major questions noticed.

1. The online distribution of surveys could already contribute bias to represent the general US population on socioeconomics, education, and other factors. Please discuss this potential bias.

2. One of the major finding of the article is recognized community exposure as predicting factor regarding the level of public knowledge and awareness of PFAS. Although a higher awareness of PFAS could increase information-seeking behavior, what is the application of this finding in promoting the general population awareness of PFAS?

3. Detailed validation would be beneficial to look into the respondent who reported “yes” to the “Has your community been exposed to PFAS?” Are they from similar communities? Has anyone reported “no” to this question from the same communities?

Reviewer #2: In this study, Berthold et al. evaluated the general public’s awareness of PFAS contamination. They found that about half of respondents had never heard of PFAS. The respondents who are aware of PFAS community exposure were more likely to have knowledge of PFAS. They conclude that PFAS information and health risks need to be better communicated to the public to help increase awareness. This study provides a general picture of public awareness of PFAS contamination, which would have effects on social and regulatory changes in the use of PFAS. I have several questions as following:

1. The authors mentioned that the quality control checks were performed by Qualtrics. Could the authors provide detailed information about how the quality control was performed. How many surveys were received, how many surveys were used in the study. If some surveys were not used, please explain the reasons. Please provide the URL of Qualtrics used in the method part.

2. In this study, the author found that almost half of the respondents had never heard of PFAS and do not know what it is. Whether it is because the areas where these people live are not contaminated by PFAS or because they lack the education on PFAS contamination?

3. For the people who are aware of community exposure to PFAS, do the authors know how they learned about this information. This can help to better promote the PFAS information.

4. It is reasonable that people who are aware of PFAS exposure are more likely to know more about PFAS. As mentioned in the study, the percentage of people who are aware of PFAS exposure but have limited knowledge about PFAS was 20%. It is important for this population to learn about PFAS since they live in the area where PFAS exposure is higher. The author should discuss this point.

5. In the discussion, could the author add several sentences about how to increase the public awareness about PFAS contamination in drinking water, especially in higher exposure area?

6. There are some typos, such as line 339 “that”.

6. PLOS authors have the option to publish the peer review history of their article (what does this mean?). If published, this will include your full peer review and any attached files.

Reviewer #1: No

Reviewer #2: No

---

## [Author Response · Author response to Decision Letter 0]

11 Sep 2023

Journal Requirements:

Authors’ response: We have updated the style of the manuscript to meet PLOS ONE’s requirements.

Authors’ response: We have made the suggested additions to clarify IRB approval methods and consent requirements. The changes are reflected in the Methods section, lines 166-171, and reads as follows: 

“The Texas A&M University Institutional Review Board (TAMU IRB) reviewed the study protocol and survey instrument prior to distribution. TAMU IRB deemed the study to be exempt from formal review. Written informed consent was obtained from all participants in the first question of the survey instrument.”

3. Please include a copy of Table 6 which you refer to in your text on page 18.

Authors’ response: The reference to Table 6 on page 18 was a typo. We have corrected the text to refer to Table 5.

4. Please include captions for your Supporting Information files at the end of your manuscript, and update any in-text citations to match accordingly. 

Authors’ response: As suggested, we have added captions for the Supporting Information files and updated the in-text citations to match. 

Comments to the Author:

Reviewer #1: 

1. The online distribution of surveys could already contribute bias to represent the general US population on socioeconomics, education, and other factors. Please discuss this potential bias.

Authors’ response: We agree that there is general inherent bias in conducting internet research, as it excludes a portion of the population without internet access from the potential sample selection. We have added the following text in the Discussion section to acknowledge this limitation:

“Additionally, the use of an online survey instrument creates inherent bias against portions of the U.S. population who may not have internet access. An estimated 7%, or nearly 23 million Americans, do not use the internet [68]. Therefore, the sample obtained from this survey may not be generalizable to this portion of the population that is not accessible via internet-based survey instruments.”

2. One of the major finding of the article is recognized community exposure as predicting factor regarding the level of public knowledge and awareness of PFAS. Although a higher awareness of PFAS could increase information-seeking behavior, what is the application of this finding in promoting the general population awareness of PFAS?

Authors’ response: The application of this finding for which methods would best raise awareness of PFAS in the general population is outside of the scope of this study. Our instrument was designed to be exploratory in nature since the scientific community has no base measure of awareness within the U.S. population. The data we collected from our respondents is meant to inform the need for future work, which could focus on the determining the best way to apply this finding. We broadly discuss potential avenues for raising public awareness with support from other studies throughout the manuscript, but we feel it is outside of the scope of our data to make definitive conclusions on how best to promote PFAS awareness. 

3. Detailed validation would be beneficial to look into the respondent who reported “yes” to the “Has your community been exposed to PFAS?” Are they from similar communities? Has anyone reported “no” to this question from the same communities?

Authors’ response: This is a good idea for a new line of inquiry with this data. However, spatial validation at this scale was not pursued in this manuscript because it is not possible to replicate multiple answers at the ZIP code scale with the sample size of 1,100 responses. Spatial analysis was not the primary goal of this study and would be better pursued in a follow-up study that addresses these details with different research questions.

Reviewer #2: In this study, Berthold et al. evaluated the general public’s awareness of PFAS contamination. They found that about half of respondents had never heard of PFAS. The respondents who are aware of PFAS community exposure were more likely to have knowledge of PFAS. They conclude that PFAS information and health risks need to be better communicated to the public to help increase awareness. This study provides a general picture of public awareness of PFAS contamination, which would have effects on social and regulatory changes in the use of PFAS. I have several questions as following:

1. The authors mentioned that the quality control checks were performed by Qualtrics. Could the authors provide detailed information about how the quality control was performed. How many surveys were received, how many surveys were used in the study. If some surveys were not used, please explain the reasons. Please provide the URL of Qualtrics used in the method part.

Authors’ response: Thank you for this suggestion. We have added statements to clarify what type of quality control checks Qualtrics uses. As stated in the manuscript, the total sample size was 1,100 surveys using a demographic quota to obtain a statistically representative sample of the general U.S. population. We cannot provide the number of surveys sent out in total, because Qualtrics did not provide that data. The URL of the Qualtrics survey could not be added because it is no longer active.

2. In this study, the author found that almost half of the respondents had never heard of PFAS and do not know what it is. Whether it is because the areas where these people live are not contaminated by PFAS or because they lack the education on PFAS contamination?

Authors’ response: This would be a good research question for a future study. What we know from our study is that people who are both unknowledgeable about PFAS and unaware of contamination in their community are unaware of PFAS overall. The survey instrument used does not delineate the cause of PFAS unawareness, therefore we believe it would be outside the scope of our study and data to attempt to address this question.

3. For the people who are aware of community exposure to PFAS, do the authors know how they learned about this information. This can help to better promote the PFAS information.

Authors’ response: Thank you for this suggestion. This would have been good additional data to collect. However, that information was not collected with this study and is beyond the scope of our analysis. This is a question that could be pursued in a follow-up study that specifically samples communities with exposure. Other studies that investigate sources of information in exposed communities are discussed in lines 133-135 of the Introduction. This is briefly revisited in lines 443-445 of the Discussion. 

4. It is reasonable that people who are aware of PFAS exposure are more likely to know more about PFAS. As mentioned in the study, the percentage of people who are aware of PFAS exposure but have limited knowledge about PFAS was 20%. It is important for this population to learn about PFAS since they live in the area where PFAS exposure is higher. The author should discuss this point.

Authors’ response: This is a great point and should be included in the text. Subsequently, we added the following statement to our manuscript in lines 446-448 of the Discussion:

“However, the results also show that 20% of the respondents who are aware of PFAS exposure in their drinking water also described themselves as having limited knowledge of PFAS. Therefore, it is important to emphasize education efforts in areas where community exposure is high”.

5. In the discussion, could the author add several sentences about how to increase the public awareness about PFAS contamination in drinking water, especially in higher exposure area?

Authors’ response: This is not within the scope of this exploratory study. However, our intention is for this manuscript to inform future research studies focused on increasing awareness of PFAS exposure. 

6. There are some typos, such as line 339 “that”.

Authors’ response: Thank you for pointing this out. We have corrected grammar and typos where needed.

---

## [Decision Letter · Decision Letter 1]

26 Oct 2023

Let's talk about PFAS: Inconsistent public awareness about PFAS and its sources in the United States

PONE-D-23-21851R1

Dear Dr. deVilleneuve,

We’re pleased to inform you that your manuscript has been judged scientifically suitable for publication and will be formally accepted for publication once it meets all outstanding technical requirements.

Kind regards,

Linglin Xie

Academic Editor

PLOS ONE

Additional Editor Comments (optional):

Reviewers' comments:

Reviewer's Responses to Questions

**Comments to the Author**

1. If the authors have adequately addressed your comments raised in a previous round of review and you feel that this manuscript is now acceptable for publication, you may indicate that here to bypass the “Comments to the Author” section, enter your conflict of interest statement in the “Confidential to Editor” section, and submit your "Accept" recommendation.

Reviewer #1: All comments have been addressed

Reviewer #2: All comments have been addressed

2. Is the manuscript technically sound, and do the data support the conclusions?

Reviewer #1: (No Response)

Reviewer #2: Yes

3. Has the statistical analysis been performed appropriately and rigorously? 

Reviewer #1: Yes

Reviewer #2: Yes

4. Have the authors made all data underlying the findings in their manuscript fully available?

Reviewer #1: Yes

Reviewer #2: (No Response)

5. Is the manuscript presented in an intelligible fashion and written in standard English?

Reviewer #1: Yes

Reviewer #2: (No Response)

6. Review Comments to the Author

Reviewer #1: All the comments have been addressed with explanation and edition. There is no further questions discovered from the manuscript.

Reviewer #2: (No Response)

7. PLOS authors have the option to publish the peer review history of their article (what does this mean?). If published, this will include your full peer review and any attached files.

Reviewer #1: No

Reviewer #2: No

---

## [Editor Report · Acceptance letter]

7 Nov 2023

PONE-D-23-21851R1 

Let’s talk about PFAS: Inconsistent public awareness about PFAS and its sources in the United States 

Dear Dr. deVilleneuve:

I'm pleased to inform you that your manuscript has been deemed suitable for publication in PLOS ONE. Congratulations! Your manuscript is now with our production department. 

Kind regards, 

on behalf of

Dr. Linglin Xie 

Academic Editor

PLOS ONE